# Impact of Air Pollution on Asthma Outcomes

**DOI:** 10.3390/ijerph17176212

**Published:** 2020-08-27

**Authors:** Angelica I. Tiotiu, Plamena Novakova, Denislava Nedeva, Herberto Jose Chong-Neto, Silviya Novakova, Paschalis Steiropoulos, Krzysztof Kowal

**Affiliations:** 1Department of Pulmonology, University Hospital of Nancy, 54395 Nancy, France; 2Development of Adaptation and Disadvantage, Cardiorespiratory Regulations and Motor Control (EA 3450 DevAH), University of Lorraine, 54395 Nancy, France; 3Clinic of Clinical Allergy, Medical University, 1000 Sofia, Bulgaria; nplamena@yahoo.com; 4Medical University Sofia, 1000 Sofia, Bulgaria; denislava.nedeva@gmail.com; 5Division of Allergy and Immunology, Department of Pediatrics, Federal University of Paraná, Curitiba 80000-000, Brazil; h.chong@uol.com.br; 6Allergy Unit, Internal Consulting Department, University Hospital “St. George”, 4000 Plovdiv, Bulgaria; novakova66@yahoo.com; 7Department of Respiratory Medicine, Medical School, Democritus University of Thrace, University General Hospital Dragana, 68100 Alexandroupolis, Greece; steiropoulos@yahoo.com; 8Department of Allergology and Internal Medicine, Medical University of Bialystok, 15-037 Bialystok, Poland; kowalkmd@umb.edu.pl

**Keywords:** outdoor air pollutants, indoor air pollutants, asthma symptoms, exacerbations, management

## Abstract

Asthma is a chronic respiratory disease characterized by variable airflow obstruction, bronchial hyperresponsiveness, and airway inflammation. Evidence suggests that air pollution has a negative impact on asthma outcomes in both adult and pediatric populations. The aim of this review is to summarize the current knowledge on the effect of various outdoor and indoor pollutants on asthma outcomes, their burden on its management, as well as to highlight the measures that could result in improved asthma outcomes. Traffic-related air pollution, nitrogen dioxide and second-hand smoking (SHS) exposures represent significant risk factors for asthma development in children. Nevertheless, a causal relation between air pollution and development of adult asthma is not clearly established. Exposure to outdoor pollutants can induce asthma symptoms, exacerbations and decreases in lung function. Active tobacco smoking is associated with poorer asthma control, while exposure to SHS increases the risk of asthma exacerbations, respiratory symptoms and healthcare utilization. Other indoor pollutants such as heating sources and molds can also negatively impact the course of asthma. Global measures, that aim to reduce exposure to air pollutants, are highly needed in order to improve the outcomes and management of adult and pediatric asthma in addition to the existing guidelines.

## 1. Introduction

Air pollution can be defined as the presence in the air of substances harmful to humans and is associated with a high risk for premature deaths due to cardio-vascular diseases (e.g., ischaemic heart disease and strokes), chronic obstructive pulmonary disease, asthma, lower respiratory infections and lung cancer [1,2]. People living in developing and overpopulated countries disproportionately experience the burden of outdoor (ambient) air pollution with 91% of the 4.2 million premature deaths in 2016 occurring in low- and middle-income countries of the South-East Asia, Central Africa and Western Pacific regions where exposure is higher [1,3]. The quality of air has been improving in the developed countries, however air pollution is steadily rising in the developing ones [4]. In order to quantify air pollution, standards of air quality for different pollutants were developed by the World Health Organization (WHO). The data from WHO indicate that 9 out of 10 people breathe air containing high levels of pollutants. More than 80% of people living in urban areas, where air pollution is monitored, are exposed to air pollutant levels that exceed WHO guideline limits. In addition, approximately 3 billion people are exposed to high levels of indoor (household) air pollution due to the use of biomass, kerosene fuels and coal for cooking and the heating of their homes, inducing a high prevalence of respiratory disorders [5].

Although there are many natural sources of air pollution such as volcanos or wildfires, it was the industrial revolution that made air pollution a real global problem [1]. Ambient air pollution affects the quality of indoor air and vice versa. According to the particle size, pollutants can be categorized as gaseous and particulate matter (PM) [6]. The main gaseous pollutants include inorganic components such as nitrogen dioxide (NO_2_), sulphur dioxide (SO_2_), ozone (O_3_), carbon monoxide (CO), carbon dioxide (CO_2_) and heavy metals such as lead or chromium (Pb or Cr), as well as volatile organic compounds (VOCs) including polycyclic aromatic hydrocarbons (PAHs). Some of them, for example NO_2_ or SO_2_, are directly produced by different pollution sources while others, i.e., O_3_ are formed by the interaction of nitric oxides and VOCs with the sunlight. The pollutants with the greatest impact on humans health are PM, which are commonly used as a measure of air quality [6,7]. Traffic-related air pollution (TRAP), a complex mixture rich in PM, exerts a particularly deleterious effect on the function of the respiratory system [5].

Asthma is a chronic inflammatory airway disease characterized by respiratory symptoms such as wheeze, dyspnoea, cough and chest tightness asssociated with variable expiratory airflow limitation. The prevalence of asthma is estimated at between 1 and 18% of the population in different countries. Evidence suggests that 13% of global incidence of asthma in children could be attributable to TRAP and data showed that air pollution has a negative impact on asthma outcomes in both adult and pediatric populations [8].

The aim of this review is to summarize the recent data about the effects of various outdoor and indoor pollutants on asthma development, symptoms, exacerbations/hospitalisations, severity, lung function and medication use, as well as to highlight the possible measures that could reduce their impact on asthma outcomes. A better knowledge of the negative impact of air pollution on asthma outcomes could help physicians (e.g., general practitioners, pulmonologists, allergologists, pediatrics, gynecologists, and emergency doctors) to improve their daily practice by adding in the interrogatory specific questions on a possible recent exposure worsening the respiratory symptoms, to educate the patients about how they could minimise the exposure and manage their asthma by an action plan. At the same time, the global awareness of air pollution effects on asthma should stimulate public health authorities and governments to take more efficient measures to limit the exposure to air pollutants.

## 2. Search Strategy, Data Sources and Selection Criteria

This review examines the literature linking air pollution and asthma across PubMed and Medline databases from January 1, 2010 and June 30, 2020. The search terms used were: “air pollution”, “outdoor air pollutants”, “indoor air pollutants”, “environmental risk factors”, “ambient sources of pollution”, “household sources of pollution”, “smoking”, and “preventive measures to reduce air pollution” associated with “asthma”. We prioritized cross-sectional and observational studies, followed by meta-analyses, systematic reviews, and general reviews. A preference was given to more recent articles published the last five years in order to have the most up-to-date evidence. Results were limited to publications in English.

## 3. Air Pollution and Risk of Asthma

The effect of air pollution on the development of asthma has been studied for many years. Increasing evidence indicates that both outdoor and indoor air pollution contributes to asthma development. Numerous cross-sectional studies provided evidence for an association between poor air quality and the incidence of asthma [9,10,11,12,13]. One of those studies, conducted in an urban population, demonstrated that the association between asthma morbidity and air pollutions was stronger in children than in adolescents and adults [10]. The important role of increased exposure to TRAP, particularly to its components PM_2.5_, PM_10_, NO_2_ and black carbon, in asthma development was demonstrated in a recent meta-analysis of 41 publications [14]. Those observations are supported by longitudinal studies evaluating the relationship between early childhood exposure to ambient air pollution and future asthma incidence. A meta-analysis of published birth cohort studies reported significant associations between long-term exposure to black carbon and PM_2.5_ and the risk of asthma in childhood up to 12 years of age [13]. The interaction between air pollution exposure in early life and asthma development was demonstrated in a prospective study on the cohort Prevention and Incidence of Asthma and Mite Allergy (PIAMA). Both early and recent exposures to PM_2.5_, PM_10_ or NO_2_, especially TRAP, were associated with a higher incidence of asthma until age of 20 years [15]. Another large population-based birth cohort study found a positive association between perinatal exposure to air pollution and asthma incidence during preschool years [16]. A recently published birth cohort study including 184,604 children born between 2004 and 2011 in Taiwan demonstrated that both prenatal and postnatal exposures to air pollutants, in particular PM_2.5_, were associated with later development of asthma [17]. The role of prenatal exposure to air pollutants in childhood asthma development was also shown in two independent meta-analyses [18,19]. The long-term effects of air pollution on asthma have been summarized in an American Thoracic Society Workshop Report, which indicates that the available evidence indicates that long-term exposure to air pollution was a cause of childhood asthma, but the evidence for a similar determinant role for adult asthma remained insufficient [20]. Some studies also provided evidence for positive associations between indoor air pollution, mainly due to cooking with polluting fuels, and asthma development in children. In a meta-analysis of 41 studies, a positive association between gas cooking, exposure to NO_2_, and childhood asthma or wheeze was found [21].

Second-hand smoking (SHS) was also reported to play an important role in asthma development. However, it is plausible that gene–environment interaction is also important for the effects of air pollution on asthma development. It has been shown recently that exposure to PM_10_ and maternal smoking was associated with a higher susceptibility for infants with an adverse genetic predisposition to asthma that also depended on the infant’s ancestry [22]. Genetic traits that affect the risk of asthma due to SHS were also demonstrated in American children participants in the Cincinnati Childhood Allergy and Air Pollution Study. Variation in the N-Acetyltransferase 1 (NAT1) gene modified asthma risk in children exposed to SHS [23]. Systematic reviews and meta-analyses have demonstrated that maternal smoking during pregnancy is a risk factor of wheezing and asthma in children, especially in the first years of life [24,25]. The mechanisms behind the adverse health effects of maternal smoking during pregnancy are still not entirely clear, but epigenetics most likely plays a role [26]. Associations between deoxyribonucleic acid (DNA) methylation at loci previously linked to in utero tobacco smoke exposure and asthma-related outcomes were observed [27]. Grandmothers smoking during pregnancy with the mother, increases the risk for asthma in the grandchild, independently of the mother’s smoking status, suggesting a transgenerational impact of prenatal tobacco smoke exposure on asthma development [28,29]. Prenatal paternal smoking exposure was also associated with childhood asthma development at 6 years of age, presumably mediated by an IgE-independent mechanism. Prenatal paternal smoking led to epigenetic modifications in certain genes as such as LIM Domain Only 2 (LMO_2_) and interleukin-10 (IL-10) via cytosine-phosphate-guanine (CpG) methylation, and these modifications are correlated to childhood asthma development [30].

Postnatal exposure to maternal and paternal smoking is also associated with wheezing in infants and pre-school children, while the data for school-aged children and adolescents are contradictory [24]. One of the limitations related to the investigation of the effect of postnatal exposure is the fact that most of the parents smoke during both the prenatal and postnatal periods, and studies on solely postnatal exposure lack consistency. A recent study showed that fathers’ smoking before the age of 15 of their children increased the risk of asthma without nasal allergies in their offspring, suggesting an effect of paternal pre-adolescent environment on the next generation [29].

Less data is available for maternal smoking and adult onset asthma. A recent study found that gestational tobacco smoke exposure is associated with new asthma diagnoses in adult offspring between 31 and 46 years, thus indicating the long-term effect of smoking. The association was accentuated in offspring who reported at age 31 as having past respiratory problems (wheeze) [31]. In addition, a reduction in forced expiratory volume in the one second (FEV_1_)/forced vital capacity (FVC) ratio was observed at age 31 years in the offspring with gestational smoke exposure [29]. Several longitudinal studies showed a positive association between active and passive smoking and the incidence of asthma in adults [32,33,34,35,36,37,38,39,40]. Women seem to be more susceptible to the effect of tobacco smoking than men [34,40]. Some of the studies suggested a stronger association between smoking and onset of asthma for non-atopics [35,37]. Other trials did not find any relation between smoking and newly onset asthma in the adult population [41,42,43]. A possible explanation of the inconsistencies between the results could be the different definitions of asthma that were used, self-reported questionnaires and evaluation, changes in tobacco smoking habits during the follow-up period and the ‘healthy smoker effect’ (reduces smoking initiation or favors smoking cessation among people more susceptible to the noxious effects of smoking) [44]. Current evidence is suggestive but not sufficient for a causal relationship between smoking and the incidence of asthma in adults and future research is needed in this domain.

## 4. Outdoor Air Pollution

The composition of outdoor air pollution is complex and dynamic. It changes from season to season, and is influenced by human activity and meteorological events [45]. Outdoor air pollutions include both primary pollutants emitted directly into the atmosphere and secondary pollutants formed in the air from chemical transformation of the primary. These chemical reactions depend on temperature and therefore can be influenced by global climate warming. Accumulated evidence suggested that air pollution cannot only aggravate asthma symptoms but might cause new-onset asthma as well. Several mechanisms have been identified and implicated. The respiratory mucosa formed by the airway epithelium represents the first contact between air pollutants and the respiratory system, functioning as a mechanical and immunologic barrier. Airway epithelial cells are connected by tight junctions and secrete mucus, host defense peptides and antioxidants, and express innate immune receptors, which could be activated by inhaled foreign substances and pathogens [45]. Under conditions of air pollution exposure, the defenses of the airway epithelium are compromised by the disruption of epithelial integrity, uptake of particles, activation of Toll-like and Nucleotid-binding Oligomerization Domain (NOD-receptors), epithelial growth factor receptor and induction of oxidative stress. Activation of these receptors results in (NF)-kB (nuclear factor Kappa B) activation, leading to pro-inflammatory cytokine expression [46]. Oxidative stress is one of the biological mechanisms proposed to partly explain the association between outdoor air pollution and asthma. Neutrophils attracted into the airways after exposure to certain pollutions produce reactive oxygen species (ROS) that induce epithelial cell inflammation, airway hyperreactivity (AHR) and lung injury. Pollutants can act directly by the production of free ROS and diffusion from the airway surface, or indirectly by inducing inflammation. Ozone (O_3_) exposure causes ROS production and changes in the expression of claudins, the major components of tight junctions, thus leading to tight junction barrier permeability and AHR [47].

Importantly, pollutants (e.g., O_3_, SO_2_) can act as adjuvants and affect the production of some cytokines (e.g., thymic stromal lymphopoetine) in airway epithelial cells, which promote T-helper 2 (Th2) phenotypic differentiation and IgE production. There is evidence for the stimulation of T-helper 17 (Th17) responses as well. Furthermore, repeated exposure to O_3_ induces group 2 innate lymphoid cells (ILC2)-mediated airway type 2 immunity and the nonatopic asthma phenotype [20]. Numerous studies have repeatedly demonstrated epidemiological links between air pollution and increased respiratory tract infections in patients of all ages, which are considered the cause of asthma exacerbations. Changes in receptor expression for pathogens, antiviral mechanisms, or host defense peptide biology could be responsible. Oxidative stress could affect intercellular adhesion molecule-1 (ICAM-1) responses to rhinoviruses in epithelial cells of the respiratory mucosa. Coexposure of airway epithelial cells to rhinoviruses and NO_2_ appears to induce a synergistic upregulation of ICAM-1, which could exaggerate the pathogen response. Available data indicated that oxidant pollutants (NO_2_ or O_3_) could amplify the generation of proinflammatory cytokines by rhinovirus-infected cells in epithelial cells of the respiratory mucosa. Oxidative stress has also been linked to a reduction in corticosteroid (CS) responsiveness in asthma patients, an important observation from a practical point of view [48].

One predisposing factor that contributes to the injury of airways by air pollutants might be atopy. Conversely, air pollutants could increase the risk of sensitization and the responses to inhaled allergen in asthma patients. Such a potential enhancing effect has been studied and demonstrated for O_3_, NO_2_, SO_2_. The mechanisms that could explain the enhanced sensitisation to aeroallergens by air pollutants include the higher deposition of allergen in the airways due to carriage by particles, an increased epithelial permeability due to oxidative stress, a greater antigenicity of proteins, and a possible direct adjuvant effect [49]. Apparently, the responses to air pollutants are diverse and individual. Genetic variations affect the function and susceptibility of epithelial cells. Specific polymorphisms in antioxidant enzyme genes, such as the glutathione-S-transferase family, especially Glutathione S-Transferase Pi 1 (GSTP1), are associated with differences in susceptibility to the adverse effects of pollutants and can modify the risk of asthmatic responses. Adults and children with Glutathione S-Transferase Mu 1 (GSTM1) null genotypes have reduced glutathione-S-transferase enzyme activity and are at increased risk of developing asthma when exposed to O_3_. The association of tumor necrosis factor (TNF) polymorphisms with asthma and differences in susceptibility to the adverse effects of pollutants has been demonstrated. TNF polymorphisms, thought to affect the expression of pro-inflammatory cytokines, seem to influence the response of the lungs to O_3_, and the risk of developing asthma [50]. Results from a recent genome-wide interaction study identified gene–NO_2_ interactions on asthma and indicated that gene–environment interactions are important for asthma development [51]. Mucin gene variants contribute to air pollutant responses in asthmatic patients. Their role in air pollution-induced mucin production has been demonstrated [52].

The impact of air pollution on asthma could by modified by other individual factors like obesity, as reported in most studies. A large cross-sectional study found that the effect of NO_2_ and SO_2_ on asthma was significantly greater in overweight or obese children. Similarly, the exposure to O_3_ is associated with a poorer lung function for obese adults when compared to people with a normal weight [53].

### 4.1. Ozone

In the troposphere, O_3_ is a secondary pollutant generated through a chemical reaction between oxides of nitrogen and Volatile Organic Compounds (VOCs) released by natural sources or following anthropogenic activities in the presence of sunlight. Other involved elements are CO and methane [54]. The combustion of fossil fuels, emissions from industrial facilities and electric utilities, gasoline vapors, motor vehicle exhaust and chemical solvents are among the main sources of O_3_ precursors. Anthropogenic emissions were responsible for 37% of O_3_ impacts in 2015 globally [55]. Due to its low water-solubility, O_3_ is not effectively removed by the upper respiratory tract and has the capacity to penetrate deeply into the lungs [7].

It is well established that inhaled O_3_ first interacts with antioxidants in the airway epithelial cells. Surfactant protein D in particular has been shown to modulate the response to O_3_ and appears to have important genetic variability that influences personal susceptibility [56]. When the dose of O_3_ in the respiratory tract exceeds the protective capacity of antioxidants, adverse health effects are likely to occur. The oxidative stress induced by the secondary oxidation results in airway inflammation, AHR, and decrements in lung function in asthmatic adults. As a highly reactive gaseous pollutant, O_3_ exerts inflammatory effects on the respiratory system. The oxygen radicals evoke oxidative stress and airway inflammation, more pronounced among allergic subjects [57].

Personal short-term exposure to O_3_ increases the risk of current asthma with persistent evidence that it could directly cause asthma exacerbation [58,59]. Increased rates of asthma hospital admissions and emergency department visits following days of elevated ambient O_3_ concentrations have been reported in some epidemiology studies. The consequences of short-term O_3_ exposure have been evaluated in meta-analysis of 47 eligible studies published recently, which confirmed the association between O_3_ exposure and asthma exacerbations measured as emergency room visits or hospitalizations. The association was significant during the warm season and in the areas where ambient O_3_ concentrations were higher [60]. As estimated recently, 9–23 million annual asthma emergency room visits globally in 2015 could be attributable to O_3_, representing 8–20% of the annual number of global visits [53]. Children appeared to be more susceptible to O_3_. This may be due to children’s higher breathing rate, narrower airways, lungs and immune system still being in development, and more frequent outdoor activities. Results from time series analysis of asthma hospital admissions and daily 8-h maximum O_3_ concentrations established significant relationships for all ages with the highest risk for children [61].

Strong evidence of a relationship between long-term O_3_ exposure and respiratory morbidity is provided by studies focused on asthma development in children and on increased respiratory symptoms in asthmatics. It was demonstrated that exposure to O_3_ in early life was significantly and positively associated with a detrimental effect on the lung function development in children, larger in boys. Nevertheless, no clear and consistent findings have been reported for the long-term effects on lung function [54]. Prenatal exposure to O_3_ has not been associated with subsequent childhood asthma [18]. O_3_ might be an important risk factor affecting the progression of asthma to chronic obstructive pulmonary disease (COPD), defining the asthma–COPD overlap (ACO) syndrome. It was established that individuals with asthma exposed to higher levels of O_3_ had greater odds of developing ACO [62]. Long-term O_3_ exposure is significantly associated with the risk of death, especially for cardiovascular and respiratory diseases. A study compared daily O_3_ concentrations to the daily number of deaths in an urban European population during 3-years. An increase in O_3_ concentration was observed during the warm period of the year, and was associated with an increase in the daily number of deaths (0.33%), notably respiratory deaths (1.13%). No effect was observed during wintertime [7].

### 4.2. Nitrogen Dioxide

NO_2_ is a traffic-related pollutant, as it is emitted from automobile motor engines. Transportation can contribute up to 80% of ambient NO_2_, so it is a convenient marker of primary pollutant. NO_2_ is an irritant of the respiratory system, which penetrates deep into the lung, inducing coughing, wheezing, dyspnea, bronchospasm, and even pulmonary edema when inhaled at high levels. It seems that concentrations of over 0.2 parts per billion (ppb) produce these adverse effects in humans, while concentrations higher than 2.0 ppb affect T-lymphocytes, particularly the CD8+ cells and natural killer (NK) cells involved in different immune responses [7]. It can augment the degree of allergic airway inflammation and prolong allergen-induced AHR. Compared with its direct effects on the airways, NO_2_ might play a more prominent role as a sensitizing agent to inhaled allergen. Exposure to 0.4 ppm NO_2_ for 4 h enhances both immediate- and late-phase responses to inhaled allergen, and can activate NF-κB, and develop allergen sensitization. High exposure to NO_2_ during the first year of life was associated with increased risk of sensitization to pollens at age 4 years [63]. Meta-analysis provided evidence for association between NO_2_ exposure during pregnancy and differential offspring DNA methylation in mitochondria-related genes. Exposure to NO_2_ was also linked to differential methylation as well as the expression of genes involved in antioxidant defense pathways [64].

NO_2_ is associated with significant morbidity in asthmatic individuals and might be a cause of incident asthma. Consistent with other studies that found associations between prior pollution exposure and future asthma risk, a recent study revealed that the odds of future asthma diagnosis for children exposed to a high concentration of NO_2_ in early life were 1.25 times greater than those for children exposed to a low concentration of NO_2_ [65]. A recent estimation on the basis of data from 194 countries concluded that, each year, 4.0 million (95% CI 1.8–5.2) new cases of paediatric asthma might be attributable to NO_2_ pollution, accounting for 13% (5.8–16) of global incidence. This contribution exceeded 20% of new asthma cases. The European analysis subset reported in the same paper estimated that 17% of the burden in Western Europe, 14% in Central Europe and 17% in Eastern Europe was attributable to NO_2_. About 92% of paediatric asthma incidence attributable to NO_2_ exposure occurred in areas with annual average NO_2_ concentrations lower than the WHO guideline of 21 ppb [66]. The longest longitudinal study of the health of school-aged children in Canada with >17 years of follow-up found that exposure to total oxidants (O_3_ and NO_2_) at birth increased the risk of developing asthma by 17% [67]. A recent meta-analysis, using observational data from five European birth cohorts, found no evidence suggesting that long-term air pollution levels including NO_2_ were associated with the prevalence of current pediatric asthma up to age eight years [68]. Compliance with the NO_2_ WHO air quality guidelines was estimated to prevent 2434 (0.4% of total cases) incident childhood asthma cases per year across eighteen European countries [11].

Studies in children and adults have identified associations between even low-levels of NO_2_ and symptoms of asthma, reduced lung function, and exacerbation of asthma. Data from several cross-sectional studies and from a meta-analysis of published studies evaluated the association between air pollution and lung function in children. NO_2_ exposure was correlated with an increase of fractional exhaled nitric oxide (FeNO) and a delayed increase in both FEV_1_ and FVC [69,70]. Even during pregnancy, NO_2_ exposure could impair lung function in early life [71]. A systematic review showed a significant association between NO_2_ exposure and moderate/severe asthma exacerbations in children and adults (OR: 1.024; 95% CI [1.005, 1.043]) [72].

### 4.3. Carbon Monoxide and Carbon Dioxide

CO and CO_2_ are produced by fossil fuel when combustion is incomplete. Higher temperatures and amounts of CO_2_ in the atmosphere are major factors that have been linked to an increased duration of pollen seasons, quantity of pollen produced by plants, and possibly allergenicity of pollen. It has been demonstrated that birch pollen extracts from trees grown in warmer temperatures had stronger IgE binding intensity. Enhanced ragweed pollen production as a function of increasing CO_2_ levels has also been established. The growth of Alternaria species can become more abundant and produce more allergens in an enhanced CO_2_ environment. These changes could affect allergic asthma. A potential link with thunderstorm-related asthma epidemics could be suspected [73].

Evidence suggests an association between exposure to CO and moderate or severe asthma exacerbations in adults (OR: 1.045; 95% CI: [1.005, 1.086]), but the link was not confirmed in children. Significant associations were observed between decreasing death rates of asthma and lower CO levels [72].

### 4.4. Sulfur Dioxide

SO_2_ is released primarily from the combustion of sulfur-containing coal and oil. People prone to allergies, especially allergic asthma, can be extremely sensitive to inhaled SO_2_. The major health problems associated with SO_2_ are bronchitis, mucus production, and bronchospasm. It is an irritant that penetrates deep into the lung where is converted into bisulfite and interacts with sensory receptors, causing bronchoconstriction [5]. In response to SO_2_, asthmatic subjects experience increased symptoms and a greater decrease in lung function at lower concentrations compared with non-asthmatics, who are often unresponsive at concentrations of less than 5 ppm. Considerable individual variations in the spirometric response to inhaled SO_2_ have been noticed, suggesting a potential genetic link. Children with a particular polymorphism in the TNF-α gene had more significant reductions in lung function after SO_2_ exposure [74,75].

A significant association between SO_2_ and both asthma prevalence and current symptoms among children, especially in those with atopy, has already been established. Decreased lung function and increased hospital admissions with enhanced SO_2_ exposure have been documented in pediatric populations. Even the low-dose SO_2_ exposure is associated with a decline in lung function (FEV_1_ and FVC) among the general population [76]. A systematic review showed a significant relationship between SO_2_ and moderate/severe asthma exacerbations in children aged 0 to 18 years (OR: 1.047; 95% CI: [1.009, 1.086]) but not in adults [72].

### 4.5. Particulate Matter

PM is a complex heterogeneous mixture of dirt, soot, smoke and liquid droplets from both natural and man-made sources. PM ambient air pollution is responsible for approximately 0.8 million premature deaths per year and 6.4 million years of life lost [77]. It was also estimated that PM_2.5_ was responsible for around 16 million incident cases of childhood asthma every year. Although particles are detected in many organs, the respiratory system is usually the first line of entry into the body. PM penetrates deeply into the lungs and increases the frequency and severity of asthma attacks, exacerbating bronchitis and other lung diseases [55].

Sources of PM could be natural or anthropogenic. The former include wind-blown dust, sea salt, volcanic ash, pollens, fungal spores, soil particles, the products of forest fires and the oxidation of biogenic reactive gases. Anthropogenic emissions of PM derive from industrial processes, construction work, mining, cigarette smoking, fossil fuel combustion and wood stove burning. The main sources of PM in the urban areas are road traffic as well as the burning of fossil fuels in power stations [74,78].

Depending on the way it is released into the environment, PM could be primary or secondary. Primary particles are introduced into the atmosphere directly from their sources (road transport, combustion as well as the land and sea through soils carried by the wind), whereas secondary PM is a product of chemical reactions among different primary particulates. The difference between these two types of PM is the length of stay in the atmosphere—it takes more time for the secondary PM to be formed and therefore its persistence is prolonged [78].

The size of PM is of vital importance since it determines, to a great extent, its impact on respiratory health and the penetration degree in the human respiratory system. According to its diameter, PM could be divided into three categories: coarse PM_10_ (from 2.5 to 10 μm), fine PM_2.5_ (from 0.1 to 2.5 μm), ultrafine PM_0.1_ (UFPs) (less than 0.1 μm). Coarse PM deposits primarily in the nasopharynx or primary bronchi; fine PM in the alveoli and terminal bronchioles; UFPs cross cell membranes and interact directly with cellular structures. The greatest number of particles fall into the ultrafine size range. UFPs have a detrimental effect on human health because their small size allows the greatest lung penetration and passage across the air–blood barrier [6,7,78].

Inhaled PM has the capacity to elicit lung oxidative stress as well as to interact with different components of the immune system and enhance allergic inflammatory response. What is more, not only do the particles infiltrate the circulatory system through layers of alveolar obstruction, but they can also absorb many airborne toxic substances on their surface, such as heavy metals, PAHs and organic/inorganic ions. It has been suggested that PM induces oxidative stress through several different mechanisms. Firstly, the redox cycle of some components of the particle’s surface, like iron or quinones, leads to the formation of ROS, hydrogen peroxide and the damaging hydroxyl radical in the lungs. Secondly, bacterial endotoxins associated with the particle surface can trigger inflammation. The particle surface itself has also been found to cause oxidative stress in vivo but this effect is not well defined yet [78]. PM enhances airway inflammation by interacting with the innate and adaptive immune system. It was suggested that PM activates neutrophils and eosinophils through increased levels of proinflammatory cytokines. PM induces antigen-presenting cell-mediated inflammatory responses as well as an imbalance of Th cells with an increase in Th2 and Th17 cells and downregulation of T-helper 1 (Th1). Exposure to PM could also lead to apoptosis and autophagy in lung epithelial cells in asthma. What is more, UFPs cross cell membranes and directly interact with cellular structures. UFPs escape the mucociliary clearance and the ingestion by alveolar macrophage scavenging [79]. A study of Mills et al. demonstrated that UFPs were detected in the blood immediately after inhalation and remained in the lungs for up to 6h after installation. Therefore, UFPs can induce severe eosinophilic inflammation, alveolar macrophage chemotaxis and epithelial damage in asthma [80].

The respiratory health effects of short-term exposure to air pollution include worsening of asthma symptoms, school absences, emergency department visits, hospitalizations and decreased lung function. Despite the limited available data, there is growing evidence about a possible impact of long-term outdoor air pollution exposures and asthma incidence. The American Thoracic Society Workshop Report revealed a strong correlation between childhood asthma and long-term air pollution exposure, especially to TRAP. PM_2.5_ was found to induce airway remodeling and an increase in the incidence/severity of asthma-like phenotypes [20]. Particulate pollution might be a risk factor for the progression of asthma to ACO. The Canadian Community Health Survey found that asthmatics exposed to higher levels of PM_2.5_ had nearly three-fold greater odds of ACO [62]. It was also demonstrated that PM increases the rate of emergency room visits due to asthma exacerbations in both adults and children. A study of Anenberg et al. concluded that 5–10 million annual asthma emergency room visits globally (4–9%) could be attributable to PM_2.5_ [55]. A significant correlation was found between increased asthma emergency room visits in adults and high PM_2.5_ concentrations, especially in the warm seasons [81]. It was also demonstrated that PM_10_ was a statistically significant risk factor for a 2% increase in the number of asthma-related emergency visits in children [82]. A recent meta-analysis revealed that UFP exposure increased the number of asthma exacerbations, subsequent emergency room visits and hospital admissions in children [83]. In contrast, the exposure to PM attributable to landscape fire smoke (PM_LFS_) seems to be associated with a higher risk for emergency department visits for the elderly compared to children. A meta-analysis showed that emergency department attendance increases with a 10μg/m^3^ PM_2.5LFS_ of 15% for elderly (95% CI [1.1–1.2]) *vs* 4% (95% CI [1–1.08]) for children [84]. The risk is greatest on the day of exposure to PM_LFS_ (an increase in emergence department attendances for asthma by 1.96% [95% CI: 0.02, 3.94]) and for women 20 years and older (5.08% 95% CI [1.76, 8.51]). [85,86].

There is growing evidence that PM exposure could be associated with impaired asthma control. A study found a correlation between poor asthma control, elevated PM_2.5_ and pollen severity in a pediatric population [87]. A cohort study of 32 asthmatic adult patients revealed that a 10 mg/m^3^ increase in PM_10_ personal exposure was associated with an increase in Saint George Respiratory Questionnaire scores and a decrease in Asthma Control Test scores [88].

PM exposure might be an important risk factor for lung function deterioration in both children and adults with asthma. An association between the fall of the FEV_1_/FVC ratio and air pollution was found in a cohort study by Yu et al. Acute exposure to PM_10_ in non-smoking adults with refractory asthma correlated with a 0.4% drop in the FEV_1_/FVC ratio in the spring [89]. A longitudinal analysis showed that an increase in PM_10_ concentration was associated with increased peak expiratory flow (PEF) variability of >20% and a decrease in the mean PEF among 64 adults with asthma [90]. It was also found that an increase of 10 μg/m^3^ in 24-h mean PM was associated with a drop of 3 L/min in PEF in children hospitalized for severe asthma exacerbations [91].

### 4.6. Outdoor Air Pollution and Asthma Outcomes

Many studies have demonstrated so far a clear association between short-term exposure to outdoor air pollutants and different asthma outcomes including asthma control [87], lung function [92], consumption of asthma medications [93,94], outpatient visits [95,96], asthma exacerbations [97,98], emergency room visits [99], hospitalizations [100,101], length of stay in the hospital [102] and deaths [2]. Several air pollutants have been implicated in the loss of asthma control. It seems that TRAP plays a particular role in this process because associations of asthma outcomes with outdoor air pollution were enhanced among subjects living in homes with high TRAP [9]. A study found that TRAP was associated with increased risk of hospitalizations due to asthma exacerbations in a population of 0–14 years of age children in California [100].

On the other hand, some air pollutants are associated with asthma morbidity independent of their source. In one study, exposure to outdoor PM_2.5_ was significantly associated with an increased number of emergency room visits due to asthma and the effect was independent of the source of PM_2.5_ [99]. An analysis of 3520 cases of acute asthma exacerbation indicated a positive association with the concentration of PM_2.5_ in the outdoor air and the association remained significant after adjusting for gaseous co-pollutants [97]. Several meta-analyses have been performed in order to single out those components responsible for asthma exacerbations. A meta-analysis of 26 studies conducted worldwide found a 4.8% increase in the risk of asthma-associated emergency department visits and admissions among children exposed to short-term increases in PM_2.5_ of 10 μg/m^3^, with greater effects in Europe and North America than in Asia [103]. In children, the association between NO_2_, SO_2_, and PM_2.5_ exposures and asthma exacerbations, as well between all outdoor pollutants and hospital admissions, were confirmed by two meta-analyses [72,104].

The effect of outdoor air pollution may change with time within the same population. In a longitudinal study, it was demonstrated that each 6.8 μg/m^3^ increase in PM_2.5_ on the same day was associated with 0.4% (0.0%, 0.8%), 0.3% (−0.2%, 0.7%), and 2.7% (1.9%, 3.5%) increases in the rate of asthma emergency department visits in the 2005–2007, 2008–2013, and 2014–2016 periods, suggesting that the toxicity from PM_2.5_ increased with the time [105].

Although the relation between exposure to outdoor air pollution and exacerbations of childhood asthma has been well documented, there is less evidence on exposure to indoor air pollution from incomplete combustion of polluting fuels. However, in a recent study, coexposure to elevated concentrations of indoor and outdoor pollutants was synergistically associated with increased emergency room visits for asthma [106].

It has been estimated that the combined effects of outdoor and indoor air pollution are responsible for approximately seven million premature deaths every year, mainly due to respiratory and cardiovascular diseases. Annually, around 500,000 deaths of children under 5 years of age and 50,000 deaths of children aged 5–15 years were attributable to air pollution. The burden of disease attributable to air pollution is not evenly distributed is greater in low- and middle-income countries than in high-income countries [107].

Table 1 shows the legal concentrations of outdoor air pollutants according to WHO guidelines and summarizes their negative impact on asthma outcomes.

### 4.7. Outdoor Air Pollution and Asthma Management

Several risk reduction measures have been recommended. These include personal strategies, community and government interventions, as well as the use of effect modifiers, which could reduce the risk factors [108]. Patients’ education to minimise their exposure to air pollutants represents an important step in asthma management. Several measures could be beneficial, like the use of close-fitting N95 facemasks when air pollution levels are high, shifting from motorised to active travel (e.g., cycling, walking), selecting low-traffic routes or those with open spaces, driving with windows closed, maintaining car air filtration systems and internal circulation, and being informed of local air pollution levels [108]. For that purpose, alerts on the occurrence of peaks of pollution, freely available online for the general population, could be helpful. This exclusion of outdoor activities during the period of poor air quality could be added to the asthma action plan. Moreover, peak pollution levels could be concomitant with exposure to seasonal aeroallergens, with an additive negative impact on asthma outcomes [109]. It was also suggested that patients with asthma ought to live at least 300 m from major roadways to reduce the impact of pollutant exposure on their asthma [49]. Community-level interventions such as urban planning of “smart” cities with more green space at distance from major traffic arteries and industrial areas, as well as the development of walking and cycling paths separated from motorised streets, may reduce respiratory morbidity [108]. Governments must monitor air pollution, inform the population about the risks when air pollution levels are high and take measures in controlling the release of PM, like considering alternative fuels such as gas, fuel-cleaning options such as coal washing as well as alternative production processes and technologies [49]. For example, an European pediatric study showed that compliance with the WHO air quality guidelines for PM_2.5_ could prevent 11% of all incident asthma cases, while the minimum air pollution levels for NO_2_ (1.5 µg·m^−3^) and PM_2.5_ (0.4 µg·m^−3^) were estimated to prevent 23% and 33% of incident cases, respectively [11].

The treatment of asthma exacerbations related to air pollution is not different from the usual clinical practice. All asthmatic patients must have a controller asthma treatment as recommended by current guidelines [8]. Inhaled corticosteroids (ICSs), the first choice treatment as a controller of asthma, proved to be beneficial in decreasing adverse responses to pollutant exposures [46]. Dietary supplements such as carotenoids, vitamin D and vitamin E are suggested to protect against airway inflammation and damage induced by pollutants that can trigger asthma initiation. Vitamin C, curcumin, choline and omega-3 fatty acids may also play a role [46,110]. Previous data showed that dietary intake of fruits and vegetables (e.g., Mediterranean diet) was associated with a better lung function, particularly among children exposed to O_3_ [46]. However, this protective effect of dietary antioxidant intake seems more evident in children with low levels of outdoor air pollution exposure, but may be insufficient for the children exposed to higher amounts of air pollutants [111].

Most of enumerated measures to reduce the impact of air pollutants on asthma outcomes are easier to apply in developed countries with adapted economical resources than in low- or middle-income countries. Poor countries often lack the technology and resources to fight pollution because their economies are still growing, so their citizens are more at risk of respiratory and cardiovascular diseases related to high levels of air pollution. Energy production is one of the most polluting activites because much of this energy comes from coal. While developed countries are more likely to invest in cleaner fuel sources, and technologies that limit emissions, governments of developing nations just want to ensure energy for their citizens at competitive and accessible prices. Even the monitoring of air pollution is sometimes difficult in developing countries; it must be encouraged, and the use of mobile devices is a less expensive solution [1,5]. In addition, evidence suggests that asthma is underdiagnosed and undertreated in low-income countries [112]. However, access to proper diagnosis and treatment with controller medications for asthma (e.g., ICSs) is feasible and cost-effective even in low-resource settings by reducing symptoms, health care utilization, mortality and improving quality of life [112]. Actions by cities and national governments are needed in developing counties to minimize the impact of air pollution on their populations’ health. Monitoring of air quality, education for health, development of healthcare systems, active and public transport infrastructure, use of better methods of energy production (e.g., renewable energy sources) and technologies to reduce emissions are efficient measures that improve air quality and consequently life expectancy and worker productivity. It is important for developing nations to find a balance between economic growth and air quality to protect the health and standard of living of their citizens [50,113].

The effects of various air pollutants on asthma outcomes and their socio-economic impact are represented in Figure 1.

## 5. Indoor Air Pollution

### 5.1. Tobacco Smoke

Tobacco smoking is the inhalation of smoke produced during the burning of tobacco leaves. Currently, more than 1.1 billion people are smokers and, additionally, another part of population is exposed to SHS. Tobacco smoke is a complex and dynamic mixture containing more than 7000 chemicals, of which at least 250 are known to be harmful and at least 69 are known to cause cancer [114]. Mainstream smoke is the aerosol drawn and inhaled directly by a smoker from a cigarette, cigar, or pipe, while sidestream smoke is the aerosol emitted in the surrounding air from a smoldering tobacco product [115]. Sidestream smoke is the main part of the SHS. The other main contributor to SHS is the exhaled portion of mainstream smoke. SHS is also known as passive smoking or environmental tobacco smoke. The aerosol of mainstream smoke is complex and consists of vapor and particulate phase. The main component of the vapor phase is the CO, but it also contains acetaldehyde, formaldehyde, acrolein, nitrogen oxides and CO_2_. Tar and nicotine form the particulate part of the mainstream smoke aerosol [116]. Tobacco smoking increases the risk of developing cardiovascular disease, stroke, COPD, lung cancer and other cancers [117]. It has an impact on asthma at various levels and it is a well known modifiable risk factor for symptom control and exacerbation. The prevalence of smoking among asthma patients is comparable to the general population (around 20%) [118].

Airway inflammation in asthmatic smokers differs from asthmatic non-smokers with higher total sputum cell counts, predominance of activated macrophages and neutrophils in sputum, airways, and lung parenchyma as in early COPD [119,120]. Previous data showed that smokers with asthma have higher sputum matrix metalloproteinase (MMP)-12 concentrations compared to non-smokers and the levels are inversely associated with lung function and positively related to sputum neutrophil counts [121]. This neutral endopeptidase is primarily responsible for the degradation of extracellular matrix components during the remodelling processes essential for normal tissue growth and repair. The excessive activity of MMPs and the impaired balance between them and their regulators, tissue inhibitors of metalloproteinases (TIMPs), have been implicated in the tissue-destructive processes associated with chronic lung diseases, including COPD and asthma [121]. In the same line, another study found reduced sputum MMP-9 activity/TIMP ratios in smokers with asthma compared with never-smokers. Low sputum ratios in asthmatic smokers were associated with persistent airflow obstruction and a reduced CT airway lumen area, which may indicate that an imbalance of MMP-9 and TIMPs contributes to structural changes to the airways in this group [121]. These results suggest that the persistent exposure to cigarette smoke drives additive or synergistic inflammatory and remodelling responses in the asthmatic airways [122]. In addition, the number of CD83+ mature DCs and B lymphocyte cells in bronchial biopsies are significantly lower in asthmatic smokers in comparison with never-smokers with asthma, which could explain the higher number of lower respiratory tract infections in the group of smokers [123]. The rhinovirus respiratory infection is an etiologic factor for severe asthma exacerbations necessitating hospitalization, and after adjustment for baseline asthma severity, rhinovirus-positive patients were more likely to be current smokers [124].

Asthmatic smokers are less sensitive to the therapeutic effects of inhaled and oral CSs in short- to medium term administration [125,126,127,128,129,130,131,132,133,134,135,136,137,138,139]. The impact of smoking in long-term CSs treatment still needs to be investigated, but the data collected showed that impairment in response to smoking was present, even after one year of treatment with ICSs, overruling the suggestion that the insensitivity could be improved with prolonged treatment [130]. Two main mechanisms explain the CSs insensitivity in smokers with asthma. The first one is the decrease in histone deacetylase-2 (HDAC-2) activity among smokers with increased inflammatory gene expression [131,132,133,134]. This is the consequence of oxidative stress. There are high levels of nitric oxide in tobacco smoke generating peroxynitrite, which leads to the inactivation of HDAC-2 via nitration and ubiquitination [135,136]. The second mechanism involves glucocorticoid receptor (GR). CSs act through the GR that has two alternative splicing isoforms, the GRα and GRβ. GRα is the classic and the functional isoform, through which the effects of CSs are mediated, while the overexpression of GRβ inhibits the action of the ligand-activated GRα [137,138,139]. The ratio of GRα/GRβ isoforms is reduced in peripheral blood mononuclear cells from current-smokers compared with non-smokers, including patients with asthma, which could lead to a lower CS [140]. These mechanisms could partially explain the poorer asthma outcomes in smokers with asthma.

Smoking patients with asthma form a separate phenotype that requires better understanding of underlying disease mechanisms and specific management.

### 5.2. Wood-Burning and Unflued Gas Heaters, Cooking Behaviors (Using Wood or Coal), Molds

Biomass combustion is known to be an important contributor to indoor air pollution in developing countries with resultant adverse health effects [141]. Residential heating devices can be a large contributor to ambient PM, notably in rural communities; the majority of these heating sources are old and inefficient, resulting in high levels of PM emissions [142]. The Consumer Product Safety Commission estimates that there are nearly 9 million wood stoves currently in use in the US [143]. Estimates of the contribution of wood-burning to ambient air quality can vary widely [144], but wood smoke accounts for 80–90% of the PM concentrations in communities with a high proportion of wood-burning households [145]. In the European Union, it is estimated that domestic woodstoves will be the dominant source of ambient PM_2.5_, accounting for 38% of all emissions by 2020 [146]. Follow-up of cohorts or panels of asthmatic patients have demonstrated that increases in levels of PM_10_ and PM_2.5_ in the indoor environment are associated with increases in severe asthma attacks, respiratory symptoms, asthma medication use, and hospital emergency department visits [147,148]. PM exposures have been shown to result in annual lung function growth deficits that are greater than those attributed to SHS exposure in children [149].

Unflued gas heaters (UFGHs) are a major source of NO_2_, nitrous acid and CO indoors, and they can also emit formaldehyde and produce water vapour [150,151]. Exposure to gas appliances or indoor NO_2_ has been associated with a worsening of asthma symptoms in children and adults. A positive association was found between indoor NO_2_ exposure and asthma exacerbations [152]. The effects of UFGHs exposure was studied in seventy-one patients with >55 years of age with mild to moderate asthma. A significant increase in respiratory symptoms was shown (wheeze and dyspnoea) when the people used UFGHs compared with days without exposure. In addition, there were significant increases in the average odds of reported wheeze and dyspnoea per hour of UFGH use. Small but significant reductions in morning to evening PEF and FEV_1_ were observed on the days when UFGH was used compared with days when other/no heating was used [153].

Cooking has been also extensively studied as a source of indoor air pollution. Most people from low- or middle-income countries use, in their homes, biomass fuel (wood, animal dung and crop residues) or coal to cook, producing high levels of indoor pollution (e.g., CO, PM). Using coal and wood as cooking fuels were identified as risk factors of asthma in child and adult populations [154,155].

Indoor sources of molds may also be a risk factor for asthma [156,157]. If more studies showed a causal relationship between indoor mould exposure and the development/exacerbations of asthma in children, a limited level of evidence was found in the adult asthma population [157]. Several species, such as Aspergillus fumigatus and versicolor, Penicillium spp or Cladosporium sphaerospermum, herbarum and cladosporioides, display a more pronounced indoor tropism [158]. Exposure to increased daily levels of basidiospores and ascospores in the first 3 months of life were associated with increased odds of wheezing among children under 24 months in a cohort study in California [159]. Fungal components are biologically active and contribute to asthma development and the severity by IgE- and non-IgE-mediated mechanisms [156]. Fungal sensitization is associated with earlier onset of asthma and demonstrates a dose-dependent relationship of symptom severity and duration [160]. The senzitization to Aspergillus fumigatus and Penicillium spp was linked to an increased risk of severe asthma [158]. Alternaria alternata sensitization is a predictor of epidemic asthma in patients with seasonal asthma and is likely to be an important factor in thunderstorm-related asthma [161]. These data suggest that indoor mold exposure can initiate asthma and influence asthma outcomes.

### 5.3. Indoor Air Pollution and Asthma Outcomes

The asthmatics who are active smokers have increased morbidity and mortality, more severe symptoms, difficulties in controlling asthma, higher rates of exacerbations, worse quality of life, and an increased number of life-threatening asthma attacks [162]. This group of patients have more frequent unscheduled doctor visits and hospital admission, thus leading to the utilization of more health care resources [8,163,164,165]. The risk of death from asthma is increased in asthmatic patients with a smoking history of more than 20 pack-years [166]. In a cohort of 147 cases of near-fatal asthma attacks, smoking was associated with a higher in-hospital and post-hospitalization mortality [167]. Several population-based surveys demonstrate that smoking is strongly associated with poorer asthma control and this seems to be dose-dependent [168].

Exposure to SHS impacts asthma control and severity in both adults and children. A systematic review and meta-analysis has shown a nearly two-fold increase in the risk of hospitalization for asthma exacerbations in children with asthma and SHS exposure [169]. Adolescents with asthma exposed to SHS were at increased risk of having a dry cough at night, wheeze and sleep disturbance due to asthma symptoms. Dyspnea and exercise limitations were also more frequent among this population. The risk of having Emergency department/urgent care visits was two times higher in asthmatic adolescents exposed to SHS compared with unexposed adolescents. The group of adolescents with asthma were also 1.5 times more likely to use a rescue medication, nebulizer treatment, or other controlling medication and over 3.5 times more likely to have had an asthma attack that required the use of an oral or injected CS [170]. A recent study found that SHS in children aggravated the severity of asthma by affecting the balance of Treg/Th17 cells with a higher percentage of Th17 cells, while the percentage of Treg cells was reduced. SHS significantly reduced the levels of FoxP3 and tumor growth factor-β, which were associated with Treg cells, and increased the levels of interleukin-17A and interleukin-23, which were associated with Th17 cells [171]. Adults with asthma who are exposed to SHS have poor symptom control, worse quality of life, lower lung function and greater healthcare utilization [172].

Asthma itself is associated with a decline in lung function over time [173] and this process is more rapid in asthmatic smokers, compared with asthmatic non-smokers [174,175,176,177]. A smoking history of ≥10 pack-years seems to be a significant predictor factor of accelerated loss of lung function [176]. Regular/former smoking reduces lung function levels with a dose–response pattern (daily smoking rate and cumulative smoking) by affecting both larger and smaller airways [177].

Smoking is a predictive factor for ACO development, a term introduced to describe patients who have features of both asthma and COPD [8]. More than 25% of asthma patients with a smoking history of at least 10 pack-years have ACO [178]. It is important to actively look for features of ACO in asthmatic smokers because these patients have more frequent exacerbations, a poor quality of life, a more rapid decline in lung function, and high mortality, compared with asthma or COPD alone [8,179].

Exposure to high levels of PM from wood-burning or NO_2_ from UFGHs is associated with more symptoms of asthma, a high rate of exacerbations and has a negative impact on lung function [147,148,152]. These effects seem more evident in children and older people with asthma [147,148,153]. Children exposed to biomass smoke from cooking have more frequent symptoms of severe asthma [180] and the risk of asthma-related symptoms is greater for males [180], while those exposed to NO_2_ from cooking with natural gas have a higher risk of asthma exacerbations [21]. A large cross-sectional study performed in China showed that adults using coal for cooking have a higher risk for asthma symptoms than those without such exposure, and they have more asthma symptoms and poorer lung function in winter than in summer [181].

Several data sources suggested an association between the sensitization to Aspergillus fumigatus or Penicillium spp. and severe asthma [158,160]. A study perfomed in Mexico City including asthmatic patients found an association between the exposure to some molds, particularly Aspergillus fumigatus, Aureobasidium pullulans, Stachybotrys chartarum, Alternaria alternata, Cladosporium cladosporioides, Cladosporium herbarum, and Epicoccum nigrum, and uncontrolled asthma in males, but not in female patients, suggesting a possible gender susceptibility [182].

The effects of indoor air pollution on asthma are summarized in Table 2.

A large cross-sectional Brazilian study including adult asthmatic patients evaluated the impact of smoking, indoor air pollution, and dual exposure on asthma outcomes. Exposure to indoor air pollution was associated with poorer asthma control, a higher proportion of severe asthma, and worsening of lung function in exposed vs. unexposed individuals. These effects were more important for the double-exposure. Exposure to indoor air pollution and double-exposure were predictive factors for uncontrolled and severe asthma in multivariate analysis [183].

All these findings suggest that indoor pollutants play a negative role on asthma outcomes, but, more importantly, an additive effect if they are associated.

### 5.4. Indoor Air Pollution and Asthma Management

The management of asthmatics exposed to indoor pollutants must include preventive measures to reduce/avoid exposure, such as smoking cessation, efficient household ventilation, use of clean fuels (e.g., methane, liquid petroleum gas, electricity, solar cookers), portable air cleaners, and pharmacologic interventions to optimize asthma treatment.

Smoking cessation must be encouraged in all possible ways to reduce the exposure of people with asthma. Smoking cessation in patients with asthma leads to better symptom control, less use of rescue medication, improved asthma quality of life score, lung function and AHR [184,185]. However, former-smokers have an accelerated FEV_1_ decline for decades after smoking cessation compared to never-smokers, but less important than for current-smokers, suggesting a lasting and progressive lung damage induced by tobacco smoke [186]. Counseling and first-line medications for smoking cessation (nicotine replacement therapy, bupropion and varenicline) significantly increase quitting rates along with increasing the chance of preventing relapses [187]. E-cigarettes could be a valid option for asthmatic patients who cannot quit smoking by other methods. A study showed significant improvements in lung function, asthma control and AHR for asthmatic patients who chose this method for smoking cessation [188]; however, e-cigarettes might generate respiratory toxicants such as acrolein, formaldehyde, and acetaldehyde [189]. A study found that current e-cigarette use is associated with 39% higher odds of self-reported asthma compared to never e-cigarette users [190]. Therefore, the recommendation of e-cigarettes as a smoking cessation tool must be balanced against the short- and long-term safety of these products.

In general, it could be suggested that the treatment of asthmatic smokers should follow the international guidelines for asthma treatment [8], but this could be challenging due to the aforementioned. Although the data shows that asthmatic smokers have decreased sensitivity to ICSs, there are studies demonstrating that long-term ICS treatment may reduce the decline in lung function in smokers with asthma, with a greather benefit for people who have smoked <5 pack-years [191,192]. The combination therapy with ICS and long-acting beta-agonist (LABA) is probably the preferable option, in asthmatic smokers, to increasing the dose of ICS, due to the relative insensitivity and the potential adverse effects of the latter [128,193,194]. One of the most significant changes in asthma management was the announcement of the Global Initiative for Asthma (GINA) 2019 recommendations regarding the use of as-required ICS/LABA as rescue medication in symptomatic mild or moderate asthma. Trials investigating this rescue medication option are lacking in asthmatic smokers; it seems reasonable that current or former-smokers should not be excluded from this new GINA recommendation [195]. Montelukast, a leukotriene receptor antagonist, represents another therapeutic option as a controller of asthma. A study showed a significant benefit of the montelukast treatment (10 mg/day) on asthma control over 6 months compared to placebo in asthmatic patients actively smoking cigarettes. This effect is comparable to the administration of 250 μg of fluticasone propionate twice daily. However, the patients with a smoking history of more than 11 pack-years experienced better benefits with montelukast than with fluticasone. The explanation of these findings is that the more intensive exposure to tobacco smoke induces an increased synthesis of leukotrienes [196]. The effect of tiotropium, an inhaled long-acting muscarinic antagonist, is comparable in current-smokers and non-smokers with asthma treated by ICSs plus a second controller, with a significant improvement of symptoms and lung function in both groups [197]. Preliminary findings suggest that biologic therapies, such as omalizumab, mepolizumab, and dupilumab, improve clinical outcomes in smokers with asthma or ACO but current data is limited [198]. The group of asthmatic smokers forms a distinct asthma phenotype with worse outcomes, altered airway inflammation and changes in the response to pharmacological treatment. Smoking cessation interventions must start as early as possible. More studies investigating the effect of pharmacological treatments in this group of asthma patients are required.

Several studies showed the positive impact of the comprehensive statewide smoke-free indoor air laws on SHS exposure and the improvement of asthma outcomes through a reduction in prevalence, respiratory symptoms and exacerbations/hospitalizations [199,200,201]. The respect of smoke-free indoor environments and public areas is beneficial, and smoke-free indoor air laws should be enforced in all states. Several studies of children with asthma in urban environments have targeted the indoor environment to improve health outcomes. Improving household ventilation by opening windows or doors, using chimneys, hoods, or exhaust fans decreased asthma symptoms in children [202,203]. The intervention strategies (e.g., education by a health coach, remediation of the exposure by air cleaners) consistently demonstrated declines in asthma-related symptoms and significant improvements in peak expiratory flow [108]. However, all of these studies were conducted in urban environments, and the nature of these exposures likely differ from that of exposures in rural environments [204,205,206]. A study showed that installing non-polluting, more effective heating (with lower levels of indoor exposure to NO_2_) in the homes of children with asthma significantly reduced respiratory symptoms, days off school, healthcare utilisation, and visits to a pharmacist [207]. Similarly, the replacement of UFGHs with high NO_2_ emission in the schools by flue gas or electric heaters reduced asthma symptoms in children during 12 weeks following the intervention [208]. In contrast, household-level interventions, such as improved-technology wood-burning appliances or air-filtration devices did not affect quality-of-life measures among children with asthma and chronic exposure to wood-smoke [209]. If the avoidance of wood-smoke or UFGHs emission is not possible, several general cautionary measures could be made in practice.

A study found that the removal of indoor molds significantly improved asthma symptoms and reduced medication use at 6 and 12 months compared with the group without this intervention [210]. The advice of a Medical Indoor Environment Counselor could be also useful to improve the quality of the indoor environment [211]. This is summarized in the following Table 3.

These measures could reduce the exposure but, unfortunately, the complete avoidance of the pollutants is impossible, so the asthma management plan and the controller medication use are mandatory for all asthmatics, as recommended by current guidelines [8].

Unfortunately, if most of these measures to reduce indoor pollution are feasible and already promoted in developed countries, their application in low- or middle-income countries is more difficult because of limited financial resources. If the use of air cleaners, low-polluting sources for cooking and heating, or personal devices to monitor indoor air pollution are less accessible for people who live in developing countries, government measures promoting health, such as smoking cessation programs and avoidance of SHS, education of people to improve household ventilation by opening windows or doors (at least during cooking periods) and mold removal, are cost-effective methods that could reduce the negative impact of indoor air pollution on asthma, and are feasible worldwide, independent of socioeconomic status.

## 6. Conclusions

Indoor and outdoor pollution represents a major public health threat with a negative impact on asthma outcomes. Current data showed that not only is the TRAP a risk factor for asthma development in children, but so are the NO_2_ and SHS exposures. A causal relation between air pollution and the development of adult asthma is not yet clearly established. The exposure to ourdoor pollutants (O_3_, NO_2_, SO_2,_ CO, PM) could induce asthma symptoms, exacerbations and hospitalizations. The effects are dose and duration-dependent. A decrease in lung function was more frequently reported for O_3,_ NO_2_, SO_2_ and PM. Active tobacco smoking is associated with poorer asthma control, more frequent exacerbations/hospitalizations, accelerated decline of lung function and a lower response to CS. Exposure to SHS increases the risk of asthma exacerbations, respiratory symptoms, healthcare utilization, and poor lung function. High-level exposures to indoor pollutants (e.g., PM from wood-burning or NO_2_ from UFGHs) could induce more symptoms of asthma and higher rates of exacerbation, and have a negative impact on lung function. The sensitization to Aspergillus fumigatus and Penicillium spp seems to be associated with more severe asthma. These negative effects are more evident in children and older people with asthma. Asthma management according to current guidelines could reduce these effects but global measures are mandatory to minimize the exposure to indoor/outdoor pollutants and to improve asthma outcomes. Limited data exists for the effects of dual exposures (e.g., tobacco smoking and other air pollutants or associations between outdoor and indoor air pollutants) on asthma outcomes. Future research is needed on double or multiple exposures, as well as in the identification of a pattern of respiratory disease that increases susceptibility to air pollution.

## Figures and Tables

**Figure 1 ijerph-17-06212-f001:**
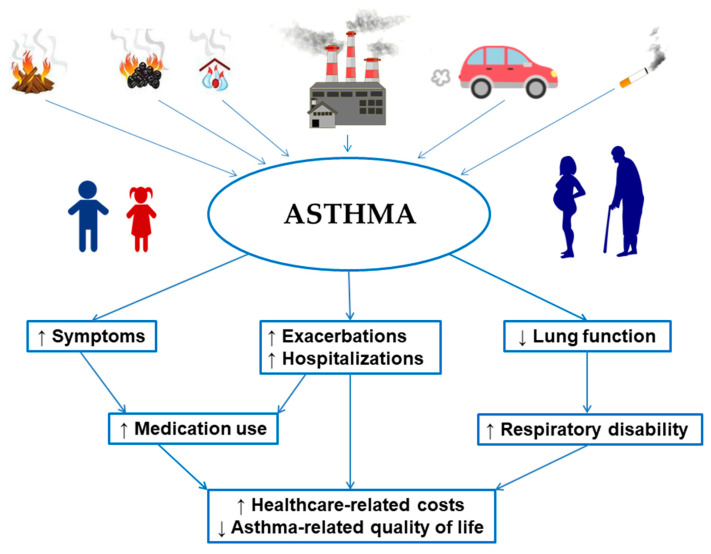
Burden of air pollutants on asthma outcomes and socio-economic impact.

**Table 1 ijerph-17-06212-t001:** Effects of outdoor air pollutants on asthma outcomes if legal concentrations are exceeded.

Pollutant	Concentration μg/m^3^	Asthma Symptoms	Exacerbations	Hospitalizations	Lung Function
**O_3_**	100 (8-h mean)	-	↑	↑	↓
**NO_2_**	200 (1-h mean)	↑	↑	↑	↓
**CO**	30 (1-h mean)	-	↑	-	-
**SO_2_**	20 (24-h mean)	↑	↑	↑	↓
**PM_2.5_**	10 (annual mean) 25 (24-h mean)	↑	↑	↑	↓
**PM_10_**	20 (annual mean) 50 (24-h mean)	↑	↑	↑	↓

**Table 2 ijerph-17-06212-t002:** Effects of indoor pollution on asthma according to their sources.

Source	Asthma Symptoms	Exacerbations	Hospitalizations	Asthma Medication Use	Lung Function
**Cigarette Smoke**					
Active Smoking	↑	↑	↑	↑	↓
SHS	↑	↑	↑	↑	↓
**Heating Sources**					
Wood	↑	↑	↑	↑	↓
Gas	↑	↑	-	-	↓
**Cooking Smoke**					
Wood	↑	-	-	-	-
Coal	↑	-	-	-	↓
**Molds**	↑	↑	-	-	-

SHS: second-hand smoking.

**Table 3 ijerph-17-06212-t003:** Recommendations to minimise indoor air pollution from heating sources.

**Recommendations to Minimise Air Pollution from Wood-Burning Heater**
Verify that your wood-burning heater conforms with standards and that the heater and chimney are installed in line with any council-specific building requirements. Burn only dry and and untreated wood. Adjust the air damper on the wood-burning heater to allow sufficient air flow to provide oxygen for clean combustion. Ensure that fresh air comes in the room to prevent carbon monoxide build up. Make sure the fire burns brightly to ensure enough heat for complete combustion. Never leave a fire smouldering overnight. Check your chimney—if there is visible smoke from it increase the airflow to the fire. Arrange for regular cleaning of the chimney.
**Recommendations to Minimise Air Pollution from** **UFGHs**
Ensure that an UFGH is the correct size for the area in your home you wish to heat. Check the electronic ignition. Have a qualified tradesperson install the gas supply system in the home. Ensure that you are aware of the instructions for the use of the appliance. Never use an UFGH overnight in the room where you sleep. Verify the proper functioning by regular inspection and maintenance of your heater.

UFGHs: Unflued gas heaters.

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
