# Peer review of "Impact of Air Pollution on Asthma Outcomes"

_ijerph, 2020, doi:10.3390/ijerph17176212_

Round 1
Reviewer 1 Report
The paper comprehensively addresses air pollution and associated health burdens. However it needs to describe the methodology section specifically to capture the exclusion/ inclusion criteria adopted in selection of cited literature. Attention to grammatical issues can greatly enhance the quality of the manuscript.

Author Response
Response to reviewers
Reviewer 1
The paper comprehensively addresses air pollution and associated health burdens. However it needs to describe the methodology section specifically to capture the exclusion/ inclusion criteria adopted in selection of cited literature. Attention to grammatical issues can greatly enhance the quality of the manuscript.
Response:
Thank you for your suggestions and advices. A specific section (no 2) was added to describe the methodology with the criteria adopted to select the studies cited in the paper and data sources.
In the abstract, we replaced NO2 by nitrogen dioxide (in yellow). On the line 44, « the » was suppressed. The sentence on the line 52 was reformulated. A coma was put on the line 59 before « as well as ». The error on the line 154 was corrected. On the line 156, « accumulating » was replaced by « accumulated ».On the line 211, « but not all » was suppressed. The sentences on the lines 212-214 and 230 were reformulated.
On the line 242, « developing lungs » was replaced by « lungs and immune system still in development ». The error on the line 248 was corrected, as well as the suggested expression on the line 249. The error on the line 266 was corrected. On the line 297, « decrease of lung function » was replaced by « reduced lung function ». On the line 300, « an » was suppressed. The sentences on the lines 301-303 were reformulated. On the line 326, « suggested » was replaced by « suggesting ». The sentence on the lines 332-333 was reformulated. The sentences on the lines 386-387 were assembled. On the line 402, « at the moment » was suppressed. On the line 403, the error was corrected.
Thank you for your suggestion for the sentence on the line 441. The modification was done.
Following your suggestion, introductory statements were added in the sections “Outdoor air pollution and asthma management” and “Indoor air pollution and asthma management” (lines 476-482, 703-706) and we used the active tense.
The sentences on lines 470-472 were reformulated.
As you suggested, the sentence on the lines 495-496 was simplified. The errors on the lines 509 and 511 were corrected.
Thank you for your suggestions concerning the sentence on the line 519. The modification was done.
As you suggested « so far » and « still » were suppressed on the lines 521 and 522. The sentence on the lines 522-523 was simplified. Following your suggestion, the paragraph from line 523 to 532 was revised and the active tense was used. « Data showed that » was suppressed from the sentence on the line 536. The word « adapted » was removed from the line 541.
The sentence on the line 560 was reformulated. The errors on the lines 567, 570, and 573 were corrected. We replaced « showed » by « has shown » on the line 592. « This » was replaced by « the » on the line 597. The sentence on the lines 607-608 was modified.
The requested information about the study cited on the lines 622-626 was added.
The paragraph from line 627 to 631 was modified into shorter sentences.
As you suggested « data » from the line 633 was replaced by « findings ».
« Should » from the line 636 was replaced by « must ». As you suggested, most of the underlined words were removed and the sentences on lines 648-651 were reformulated.
The sentences on the lines 653-657 were removed.
As you suggested, the paragraph from line 665 to line 684 was revised.
The sentence on lines 705-707 was modified. The word « check » in the ex-Table 2 was replaced sometimes by « verify ».
The sentence on the lines 728-730 was reformulated.
Reviewer 2 Report
This is very important narrative review focusing on air pollution and asthma. The author correctly addressed the effect of air pollution and asthma in children, second-hand smoking, postnatal exposure, parental smoking and maternal smoking on asthma development. I missed a discussion on occupation, fossil fuel and smoke from bushfire exposure on asthma.
Author Response
Reviewer 2
This is very important narrative review focusing on air pollution and asthma. The author correctly addressed the effect of air pollution and asthma in children, second-hand smoking, postnatal exposure, parental smoking and maternal smoking on asthma development. I missed a discussion on occupation, fossil fuel and smoke from bushfire exposure on asthma.
Response
Thank you for your suggestions. We did not discuss the occupational exposure because another paper (already published online) in the same number of the journal is focused on this subject (Tiotiu et al, “Progress in occupational asthma”, Int J Environ Res Public Health. 2020 Jun 24;17(12):4553. doi: 10.3390/ijerph17124553). Following your suggestion, we added a paragraph about the impact of bushfire exposure on asthma in the section dedicated to PM exposure (outdoor pollution, lines 413-419) and a paragraph about the impact on asthma of the smoke produced during cooking period by different energetic sources (indoor pollution, lines 678-683), as well as the measures able to reduce these exposures (lines 755-757).
Reviewer 3 Report
This is a well-written manuscript about a critically important topic. The authors performed literature reviews on air pollutants and asthma. However, the following suggestions may help to improve the manuscript even further:
- This to summarize the current knowledge on the effect of various outdoor and indoor pollutants on asthma outcome, but current knowledge is from where and the period, why did you choose these reference? You should address more detail.
- for air pollutants burden on asthma management, you can summarize the current asthma management as tables or figures, that readers can clearly know the current air pollutants burden on asthma management.
- there are a variety of definition of asthma from different studies , how to select in the this review?
- developed area and developing area are different, you should discuss separately.
- I think that policymakers who read this manuscript will want to know if the data are robust enough to be the foundation for country-wide policy development. I think it would strength the manuscript to explain what additional information is needed to support stronger air pollutant policy.
Author Response
Reviewer 3
This is a well-written manuscript about a critically important topic. The authors performed literature reviews on air pollutants and asthma. However, the following suggestions may help to improve the manuscript even further:
Response
Thank you for your appreciation and your suggestions.
This to summarize the current knowledge on the effect of various outdoor and indoor pollutants on asthma outcome, but current knowledge is from where and the period, why did you choose these reference? You should address more detail.
A section “Search strategy, data sources and selection criteria” was added where we explain the databases used, the keywords, the selection criteria for the papers and the period.
for air pollutants burden on asthma management, you can summarize the current asthma management as tables or figures, that readers can clearly know the current air pollutants burden on asthma management.
Table 1 in the paper summarizes the effects of outdoor air pollutants on asthma outcomes. Another table (Table 2) was done to show the effects of indoor air pollutants on asthma. A figure was added to show the impact the current air pollutants burden on asthma management. We hope that now the message is clearer for the reader.
there are a variety of definition of asthma from different studies, how to select in the this review?
We used the definition from GINA guidelines. (Reference 6)
developed area and developing area are different, you should discuss separately.
Thank you for your suggestion. We revised the sections “Outdoor air pollution and asthma management”, respectively “Indoor air pollution and asthma management” and added paragraphs to highlight the differences in the management between developed and developing countries according to their economic resources (lines 510-529, 778-786).
I think that policymakers who read this manuscript will want to know if the data are robust enough to be the foundation for country-wide policy development. I think it would strength the manuscript to explain what additional information is needed to support stronger air pollutant policy.
This is a general review and not a systematic review or a meta-analysis so, in our opinion, it is difficult to pretend that this paper could be a “foundation for country-wide policy development”.
Reviewer 4 Report
General comments
The major strength of this paper is that it aggregates multiple studies to highlight the most recent evidence about the association between air pollution and asthma outcomes. Overall, I recognize there is a substantial work behind this literature review but for being a systematic review, the rationale and the selection procedures should be included. Specifically, the review’s objectives should be better defined and the study designs more appropriate for answering the specific questions should be briefly described. The introduction appears unfocussed and does not cover well the existing evidence.
Some concerns for improvement of the paper are as follows.
- Introduction is very hard to follow the main focus of the study. First paragraph (lines 39-53) seems to build the background of the study, but the main topic of the study is not clear yet. The second paragraph (lines 54−69) is too redundant and unclear why the authors mention the sources of air pollutants. Authors may want to rewrite first and second paragraphs and consider sticking to the issues of air pollution and asthma relevant to their study objectives. The authors also need to address why specifically this review is needed as there remain some reviews evaluating the relationship between air pollution and asthma.
- The research strategy is not included in the original manuscript. I suggest to add methodological issue after introduction. This section should include the study design more suitable to your research questions. The inclusion/exclusion criteria for studies regarding each topic of interest should be included. The type of studies selected should reflect your research question.
- Authors may want to provide the summary of recent studies based on their main topics in table. For each topic the number of studies selected and a brief description of them should be reported (i.e. the number of time series, cross-sectional, or cohort studies, the number of studies from the US, Europe, other countries).
- Section 2.1 (outdoor air pollution): Smoking is probably the most important confounder for the association between asthma and exposure to air pollutants. The authors should discuss how the reviewed studies have dealt with this.
- The discussion about possible limitation of the present review should be included.
- Conclusions should be revised: authors should point out the topics where your review has identified there is contrasting or limited evidence, and focus on the possible future research areas. I advise that an important point for future research is the identification of a pattern of respiratory disease that increases susceptibility to air pollution.
Author Response
General comments
The major strength of this paper is that it aggregates multiple studies to highlight the most recent evidence about the association between air pollution and asthma outcomes. Overall, I recognize there is a substantial work behind this literature review but for being a systematic review, the rationale and the selection procedures should be included. Specifically, the review’s objectives should be better defined and the study designs more appropriate for answering the specific questions should be briefly described. The introduction appears unfocussed and does not cover well the existing evidence.
Response
Thank you for your appreciation and your constructive suggestions. In our opinion, this paper is not a systematic review (we don’t have the criteria requested by a systematic review https://doi.org/10.1016/j.acuroe.2018.07.002) but a general review which could explain why we did not make a Table with the number of studies selected, their description and the most important findings, neither a section with the limitations.
Some concerns for improvement of the paper are as follows.
Introduction is very hard to follow the main focus of the study. First paragraph (lines 39-53) seems to build the background of the study, but the main topic of the study is not clear yet. The second paragraph (lines 54−69) is too redundant and unclear why the authors mention the sources of air pollutants. Authors may want to rewrite first and second paragraphs and consider sticking to the issues of air pollution and asthma relevant to their study objectives. The authors also need to address why specifically this review is needed as there remain some reviews evaluating the relationship between air pollution and asthma.
R: The introduction was modified and the objectives better defined. A paragraph about the importance of this topic was added (lines 39-82).
The research strategy is not included in the original manuscript. I suggest to add methodological issue after introduction. This section should include the study design more suitable to your research questions. The inclusion/exclusion criteria for studies regarding each topic of interest should be included. The type of studies selected should reflect your research question.
R: Following your suggestion and from the others reviewers (no 1 and 3) we added a section with the “Search strategy, data sources and selection criteria) (lines 83-91).
Authors may want to provide the summary of recent studies based on their main topics in table. For each topic the number of studies selected and a brief description of them should be reported (i.e. the number of time series, cross-sectional, or cohort studies, the number of studies from the US, Europe, other countries).
R: Thank you for your suggestion. We already did two tables to summarize the effects of various air pollutants on asthma. In our opinion the table which you suggest is mandatory for a systematic review or a meta-analysis but our paper is just a general review.
Section 2.1 (outdoor air pollution): Smoking is probably the most important confounder for the association between asthma and exposure to air pollutants. The authors should discuss how the reviewed studies have dealt with this.
R: We agree with you. Unfortunately, most of studies on air pollutants don’t take in account the smoking. The most interesting study which compares the dual exposure to one exposure was already cited ((lines 694-699, reference 186).
The discussion about possible limitation of the present review should be included.
R: The most important limitation of this paper is that it is a general review and not a systematic review. In our opinion, putting the possible limitation is mandatory for a systematic review but not really necessary in a general review because its limits are already recognized.
Conclusions should be revised: authors should point out the topics where your review has identified there is contrasting or limited evidence, and focus on the possible future research areas. I advise that an important point for future research is the identification of a pattern of respiratory disease that increases susceptibility to air pollution.
R: Thank you for your suggestions. According your advice, we revised the “Conclusions” (lines 801-807).
Round 2
Reviewer 4 Report
The authors have addressed my concerns.